# Design of Ag/TiO_2_/Ag Composite Nano-Array Structure with Adjustable SERS-Activity

**DOI:** 10.3390/ma15207311

**Published:** 2022-10-19

**Authors:** Xiaoyu Zhao, Wei Xu, Xiuxia Tang, Jiahong Wen, Yaxin Wang

**Affiliations:** 1College of Materials and Environmental Engineering, Hangzhou Dianzi University, Hangzhou 310018, China; 2The College of Electronics and Information, Hangzhou Dianzi University, Hangzhou 310018, China; 3Shangyu Institute of Science and Engineering, Shaoxing 312000, China

**Keywords:** tunable nanostructures arrays, Ag film, TiO_2_ film, nano-sphere etching (NSE), hot spots, surface-enhanced Raman scattering (SERS)

## Abstract

How to fabricate large area controllable surface-enhanced Raman scattering (SERS) active nanostructure substrates has always been one of the important issues in the development of nanostructure devices. In this paper, nano-etching technology and magnetron sputtering technology are combined to prepare nanostructure substrate with evolvable structure, and Ag/TiO_2_/Ag composites are introduced into the evolvable composite structure. The activity of SERS is further enhanced by the combination of TiO_2_ and Ag and the electron transfer characteristics of TiO_2_ itself. Deposition, plasma etching, and transfer are carried out on self-assembled 200 nm polystyrene (PS) colloidal sphere arrays. Due to the shadow effect between colloidal spheres and the size of metal particles introduced by deposition, a series of Ag/TiO_2_/Ag nanostructure arrays with adjustable nanostructure substrates such as nano-cap (NC), nano cap-star (NCS), and nano particle-disk (NPD) can be obtained. These nanoarrays with rough surfaces and different evolutionary structures can uninterruptedly regulate optical plasmon resonance and reconstruct SERS hotspots over a large range, which has potential application value in surface science, chemical detection, nanometer photonics, and so on.

## 1. Introduction

Surface-enhanced Raman scattering (SERS) has been widely used since it was discovered in 1974 because of its non-destructive, high sensitivity, and high selectivity [1,2]. It has broad and great potential for applications in many fields such as biochemistry [3,4,5], chemical monitoring [5,6], surface science [6], and elucidation of the structural dynamics of molecular transformations [7]. There are many methods available to fabricate SERS substrates with high reproducibility and excellent performance [8,9]. The excellent performance of large area dense Raman signals that can enhance the local surface of the substrate makes these substrates more attractive [10,11]. Since the process of preparing nanostructures will be accompanied by the clusters of metal nanoparticles, these clusters will reduce the density of hot spots and the degree of order of hot spots after a certain period of time, so the enhancement factor (EF) will also decrease [12,13,14]. Therefore, how to fabricate structures with both large-area hot spots and high-density hot spots on SERS substrates is still worthwhile research, and low-cost nanosphere etching (NSE) just needs a little adjustment to meet this requirement [4,6,15]. Many highly ordered nanoarray patterns can be fabricated including periodic nanocaps array [16,17], nanotriangles array [18,19,20], nano dishes array [21,22], nanotubes array, nano rings array [23,24], and nanopillars array [25]. By adjusting the corresponding experimental parameters, more nanogaps can be formed in the above nanostructures [16,17,18,19,20,21,22,23,24,25]. Therefore, it can significantly increase the hot spot density in the nanostructures prepared by NSE technology [18,20].

When deposit metal materials on the colloidal sphere array template by magnetron sputtering, two types of complementary nanostructure arrays are formed on the substrate: one is the NC array formed by the metal film, the other is the hexagonal star array left on the substrate after the template is removed [25,26]. Precious metals with local surface plasmon resonance (LSPR) properties are usually deposited on the template to enhance the local electromagnetic field, while other materials, such as nano-porous GaN [27,28], Si [29,30] and ZnO [30], are also used in this process because of their corresponding properties. TiO_2_ is often used as an excellent carrier material because of its special optical [31,32] and chemical properties, so it is one of several widely used wide band gap semiconductors [33,34]. In this paper, a simple and efficient method for the preparation of nanostructure arrays is proposed by using the complementary method of plasma etching and magnetron sputtering and three kinds of nanocomposite structures with different hot spots are prepared. On this basis, TiO_2_ is introduced to further adjust the performance of SERS from the point of view of electron transfer. With the introduction of TiO_2_, the Raman enhancement factor of SERS reached 4.93 × 10^5^, which is more than twice as much as other similar work [35,36,37,38]. Through the control of the above conditions, the position and distribution of hot spots can be effectively adjusted to significantly increase the density of hot spots.

## 2. Materials and Methods

### 2.1. Preparation of Large Area Ordered PS Colloidal Sphere Array

The diameter of polystyrene colloidal sphere is 200 nm, the size deviation is less than 10%, and the solution with a concentration of 10 wt% is purchased from Duke Co (Durham, NC, USA). The deionized water (18.2 MΩ·cm^−1^) is prepared by a Millipore water purification system. We purchased Ag and TiO_2_ target (99.99 wt%, Beijing, China) from Beijing TIANRY Science and Technology Developing Center. According to our previous work, dense PS monolayers are prepared by self-assembly method. The wafer is boiled with ammonia, hydrogen peroxide, and deionized water at a volume ratio of 1:2:6 for 20 min to enhance the hydrophilicity of the wafer surface. The 100 μL polystyrene bead solution is mixed with 70 μL anhydrous ethanol by ultrasonic 3 min, the solution is dispersed on the deionized water surface, and then the liquid surface is acted with air. The colloidal spheres are self-assembled to form a dense monolayer, and finally the monolayer is extracted onto the silicon wafer [39].

### 2.2. Fabrication of the Ag/TiO_2_/Ag Nano Structural Arrays

Ag and TiO_2_ films are prepared on PS colloidal sphere array by magnetron sputtering (ATC 1800-F, USA AJA). In the process of thin film deposition, the distance between the PS colloidal sphere array and the target is 20 cm. The bottom pressure of the vacuum chamber is 4.5 × 10^−4^ Pa. Working gas is Ar with purity of 99.999% and the pressure is 0.6 pa. The sputtering power of 60 W is applied to the target TiO_2_, and the sputtering power of 10 W is applied to the target Ag. The Ag NC array peeled off by double-sided tape is then inverted and transferred to another clean silicon wafer. Then, the PS colloid spheres are etched by different times (0 s, 60 s, 120 s, 180 s, and 240 s) with a rate of 1 nm s^−1^ in a plasma cleaner equipment (RIE–10NR). With the increase of time, the diameter of the colloidal sphere is etched smaller and smaller, but the order of the array does not change. Finally, 5 nm thick TiO_2_ films and 20 nm Ag films are deposited on the etched PS colloidal microsphere array under the same deposition conditions, i.e., etched after 0 s, 60 s, 120 s, 180 s, and 240 s. With the increase of etching time, the nanostructure gradually evolves from NC to NCS and NPD.

### 2.3. Characterization of the Nano Structural Arrays

The morphology and microstructure are measured on scanning electron microscope (SEM, 15 kV, JEOL 6500F). Moreover, 4-mercaptobenzoic acid (4-MBA) is purchased from Sigma-Aldrich Co Ltd. X-ray diffraction (XRD is performed on the Rigaku D/MAX 3C X-ray diffractometer with Cu Kα radiation (λ = 1.5418 Å)). The UV-Vis spectrum is obtained by Shimadzu UV-3600 spectrophotometer (Kyoto, Japan). The Raman spectrum is collected on Renishaw Raman with a spectral resolution of 1 cm^−1^ (London, UK, 2000 confocal microscope spectrometer). The Raman laser wavelength is 785 nm, the power is 500 mw, and the single acquisition time is 1000 ms.

### 2.4. Finite-Difference Time-Domain (FDTD) Simulations

The electric field distributions of different nanostructure arrays are analyzed using commercial software (Lumerical FDTD solution). During the simulation, the substrate is on the x-y plane, and the simulated nanostructures have corresponding boundary conditions in all three directions of the x, y, and z [40]. The laser of 785 nm is used as the simulated excitation light source, the direction is vertical incidence, and the refractive index of the colloidal sphere is 1.585. The dielectric constant and refractive index of other materials are obtained from the corresponding material database [41]. The geometric parameters of nanostructures are obtained by calculation and SEM images.

## 3. Results and Discussion

### 3.1. The Preparation Process of the Ag/TiO_2_/Ag Nano Structural Arrays

Figure 1 shows the flow chart of the preparation of Ag/TiO_2_/Ag nanostructure arrays. Firstly, an ordered polystyrene colloidal sphere array (200 nm) is arranged on the silicon wafer (Figure 1A). The method of arranging the colloidal spheres is self-assembly technology, which has been mentioned in previous studies in the group. Then, a layer of Ag film is vertically deposited on the ordered PS colloidal sphere array by magnetron sputtering to form a NC array (Figure 1B). Then, the tightly arranged NC are stripped from the silicon wafer with double-sided tape and inverted and transferred to another silicon wafer (Figure 1C). When the tightly packed NC are completely inverted, some Ag nano bowl (NB) arrays are formed. Then, the PS colloidal spheres in the Ag NB are etched at different times (0 s, 60 s, 120 s, 180 s, and 240 s) to form smaller colloidal spheres. Finally, using the same deposition conditions, 5 nm thick TiO_2_ films and 20 nm Ag films are deposited successively on the etched PS colloidal sphere array. Several different nanostructure arrays can be obtained by this method (Figure 1D–F).

Figure 2 shows the SEM image of the Ag/TiO_2_/Ag NC array etched after 0 s, 60 s, 120 s, 180 s, and 240 s. With the increase of etching time, the nanostructure gradually evolves from NC to NCS and NPD. Due to the increase of particle size after the introduction of TiO_2_, different nanostructures are formed under the combined action of particle size and interstitial size. Here, we can calculate the critical value of the process from NCS to nano cap-ring (NCR). From Figure 2G, it can be concluded that when the diameter of colloidal sphere D gradually decreases from 200 nm to 140 nm, the distance between the tips of NCS d_tt_ gradually change from 70 nm to 10 nm, with the further decrease of sphere diameter, which decreases from 140 nm to 80 nm, and the distance between vertices of NCS d_tt_ gradually coincident from 10 nm to −10 nm. These calculated results are the same as the experimental results such as Figure 2B (d_tt_ = 70 ± 5 nm and D = 200 ± 10 nm), Figure 2C (d_tt_ = 10 ± 5 nm and D = 140 ± 10 nm), and Figure 2D (d_tt_ = −10 ± 5 nm and D = 80 ± 10 nm). Figure 2B–F show SEM images of four types of Ag/TiO_2_/Ag nanostructure arrays. Because the diameter of the etched PS colloidal sphere is different, the reflection color of the array structure on the substrate is also different, as shown in the corresponding optical photo in the figure. For nanostructures with the same diameter and shape, a uniform and consistent reflection color represents a high degree of order on the array. There are two main factors affecting these Ag nanostructure arrays: one is the time of plasma etching, and the other is the particle size of TiO_2_ deposited by magnetron sputtering. When the etching time is 0 s, there is still a NC array formed by deposition on the substrate. The PS colloidal spheres are closely adjacent to each other, the triangular area between the spheres is too small, and the introduced TiO_2_ particle size is larger than this region size, so an effective nano-pattern cannot be formed, as shown in Figure 2B. When etching for 60 s, due to the increase of the gap between the spheres, some NCS with an average side length of 95 ± 5 nm appear in the gap region, and because the direct distance between the spheres is still not enough to accommodate the entry of TiO_2_ particles, so the ring pattern cannot appear, as shown in Figure 2C, the average distance between the tips of the adjacent NCS is d_tt_ = 10 ± 5 nm. When the etching time is further increased to 120 s, due to the further increase of the linear distance between the spheres, TiO_2_ particles can enter the gap, so the adjacent nano-triangles cover each other, forming the pattern of NCR as Figure 2D show. When etching 180 s, the distance between the colloidal spheres is very large, forming an NPD structure, as shown in Figure 2E. When etching 240 s, as Figure 2F show, the nanostructure has not changed, but the PS colloid sphere is completely etched. XRD patterns in Figure 2A show the existences of Ag and TiO_2_ in the composite. Ag diffraction peaks at 38.1°, 44.3°, and 64.5° can be assigned to the face-centered cubic crystal structure (111), (200), and (211), in agreement with JCPDS card 04-0783. The diffraction peaks at 33.6° and 51.1° agree with TiO_2_ (211) and (411) in JCPDS card 33-1381, indicating TiO_2_ in rutile phase.

### 3.2. Evaluation of SERS Activity of the Ag/TiO_2_/Ag Nano Structural Arrays

In the UV-Vis spectrum of nanostructures (Figure 3A), different absorption bands can be observed by changing the introduction of different materials. The absorption peak around 300–400 nm is related to the absorption of TiO_2_-Ag band gap, while the change of absorption peak produced at 400–800 nm can be related to Ag-induced surface plasmon resonance (SPR) [34,42,43]. It can be observed that the absorption peak will blue shift with the change of the structure (Figure 3A), This may be because the Ag and TiO_2_ binding in the nanostructure reconstructs the Fermi level as the structure changes [44,45,46]. The increase and then decrease of resonance intensity can be attributed to the increase of nano-gap with the increase of etching time from NC to NCS and NPD. The number of hot spots gradually increases, which strengthens the formant due to the surface plasmon, and then with the further prolongation of the etching time, the gap is too large to produce effective hot spots, and the formant gradually weakens. It can be concluded from the first three absorption spectra that the resonance intensity of the NCR is the strongest and the bandwidth is the widest. According to literature reports, this may be due to a lot of electrons from the unique optical properties of TiO_2_, thus reconstructing a reconstructed Fermi level in the Ag/TiO_2_/Ag nanostructure [45,46]. In addition, since the interband absorption threshold energy of Ag starts from hv = 3.9 eV, the absorption spectrum shows a sharp increase in the spectral line at 320 nm [43]. By regulating different composite structures, the absorption peak can be adjusted within the 400–800 nm range, which is wider than that of the same type of work [47,48,49].

We choose the 785 nm laser as the excitation light to further verify the conclusion of the resonance intensity of plasmon on the surface of different structures in the absorption spectrum by Raman spectroscopy. As shown in Figure 3B, the SERS spectrum of the Ag/TiO_2_/Ag nanostructure array, where we use 4-MBA to be SERS probe. There are two strong surface-enhanced Raman scattering peaks that appeared at 1575 cm^−1^ and 1073 cm^−1^, corresponding to aromatic ring vibration, respectively, which are used to evaluate the surface-enhanced Raman scattering intensities for different nanostructures. The peak intensities of C-H deformation and COO^−^ stretching vibration modes are weak, which are reflected by 1173 cm^−1^ and 1357 cm^−1^, respectively, in Raman spectra [50,51,52]. For Ag/TiO_2_/Ag nanostructure arrays, the SERS intensity of 4-MBA is positively correlated with etching time; with the increase of the time, the intensity reaches the maximum at 120 s, then decreases gradually, and decreases significantly at 240 s. On the one hand, the intensity of surface-enhanced Raman scattering is affected by the local enhancement of the electromagnetic field caused by the LSPR effect of the precious metal; on the other hand, it is also related to a large number of electron transfer provided by TiO_2_ in our structure. Therefore, the phenomenon of Raman intensity change can be attributed to the increase of the number and size of nano-gaps with the increase of etching time from 0 s to 120 s, resulting in an increase in hot spot density. At the same time, the increased nano-gap makes it more convenient for TiO_2_ particles to enter and provide a large number of electrons associated with Ag, resulting in a gradually increased SERS enhancement, but when the etching time continues to increase, the nano-gap size expands beyond the near-field range, so hot spots decrease and the SERS signal decreased. According to the calculation, the EF of different array structures is calculated to be: 5.21 × 10^4^, 1.84 × 10^5^, 4.93 × 10^5^, and 3.12 × 10^5^. Among them, the EF of Ag/TiO_2_/Ag NC (nano cap-ring) array is the highest, which is slightly higher than that of the same type of SERS substrate.

Since we speculate that the change of SERS signal is due to the interaction between the chemical absorption of 4MBA molecules and nanostructures, the charge transfer (*CT*) of Ag/TiO_2_/Ag nanostructures is calculated, and the results are shown in Figure 3C. The degree of charge transfer is calculated by using the 1073 cm^−1^ (*b*_2_) peak belonging to the CCC in-plane bending and C-S stretching combination band (δ_CCC_ + v_CS_) and the 1575 cm^−1^ (*a*_1_) peak belonging to the benzene ring (v_CC_) [53]. The results show that the relative intensity ratio of the two Raman peaks of 4-MBA has changed significantly. The charge transfer of Ag/TiO_2_/Ag nanostructures increases at first and then decreases.

According to the theory of *CT* (*ρCT*) degree proposed by Lombardi et al., we do the same calculation [54,55]. To quantitatively calculate the relationship between *CT* resonance and SERS intensity, the *ρCT*(*k*) satisfies the following equation:(1)$$\rho CT\left(k\right)=\frac{I^{k}\left(CT\right)-I^{k}\left(SPR\right)}{I^{k}\left(CT\right)+I^{0}\left(SPR\right)},\;$$

Here, *k* is to identify individual molecular lines in the Raman spectrum. Two peaks (1073 cm^−1^ and 1575 cm^−1^) are chosen to simplify the calculation. Regarding the 1575 cm^−1^ (a_1_), its signal contribution comes entirely from SPR, and its intensity is expressed by $I^{0}\left(SPR\right)$, and for this line $I^{k}\left(SPR\right)=I^{0}\left(SPR\right)$. The other peak, 1073 cm^−1^ (*b*_2_), is asymmetric, and its strength contribution is not limited to SPR (its intensity is expressed by $I^{k}\left(CT\right)$) [56]. It represents the intensity part of the SERS intensity contributed by the CT resonance, which is independent of the intensity contributed by the surface plasmon resonance. So, $I^{k}\left(SPR\right)$ is normally to be zero. Equation (1) can be reduced to as follows:(2)$$\rho CT=\frac{\frac{b_{2}}{a_{1}}}{1+\frac{b_{2}}{a_{1}}},$$

Figure 3C show the change of relative peak intensities of I_1073_/I_1575_ vibrational mode bands. It also reflects the change of CT transition between the lowest empty orbitals (LUMO) of molecules and the Fermi levels reconstructed by different nanostructures [57,58]. The gap between TiO_2_ nanostructures and Ag nanostructures increases from 0 s to 120 s, which is more beneficial to the mixing and interface diffusion between TiO_2_ and Ag, so the mixture components of TiO_2_ and Ag are formed [46]. The special optoelectronic properties of TiO_2_ make it produce a new Fermi level in the nano structure, which leads to the increase of the transfer degree of a large number of transferable electrons between the structure and 4-MBA molecules, while with the increase of etching time, the nano-gap breaks through a certain size, resulting in Ag can completely cover TiO_2_, so it is impossible to effectively form composites and play the role of TiO_2_, so the degree of charge transfer is gradually weakened. The magnitude of CT (ρ CT) reflects the difference between the Fermi level and the molecular LUMO energy level, as well as the matching degree of the excitation energy of the Ag/TiO_2_/Ag composites, which indicates that the b_2_ mode of 4-MBA molecule is partially enhanced by CT resonance transition [59].

### 3.3. Finite-Difference Time-Domain (FDTD) Simulations of the Ag/TiO_2_/Ag Nano Structural Arrays

In addition to charge transfer, another important factor affecting the surface enhancement effect of Ag/TiO_2_/Ag nanostructures is the change of nanostructures. The FDTD numerical simulation method is used to calculate the electromagnetic field around the Ag/TiO_2_/Ag array composed of NC, NCS, NCR, and NPD, and the influence of different nanostructures on the hot spot intensity is studied. The results show that different morphologies have obvious influence on the distribution of hot spots [56,59,60]. For NC arrays, the hot spot is located in the area between the nearest two NC, as shown in Figure 4A. For the NCS array, because a large number of NCS fill the nano-gap, the hot spot density increases significantly, as shown in Figure 4B. When the structure further evolves and the distance between the NCS is reduced to a certain size to become NCR, the hot spot density further increases, as shown in Figure 4C, when the hot spot density and strength reach the strongest. In the simulated side view of Figure 4C, we can see the enhanced position of the electromagnetic field, which shows that the surface of the Ag NCR is enhanced. Therefore, it can be concluded that the hybrid array of NCR has the highest hot spot density. For the NPD array, the simulation shows that the strong electromagnetic field is only distributed in a small part of the bowl bottom, while the electromagnetic field in other places is very similar. As shown in the figure above and below in Figure 4D, the size of the nano-gap is too large. The sharp decrease of hot spot density leads to relatively weak surface-enhanced Raman scattering signals. The simulation results of the above four types of nanostructure arrays are the same as the experimental data, indicating their feasibility in simulating the distribution of hot spots.

## 4. Conclusions

In this paper, a simple and efficient preparation method is proposed to prepare large area composite nanostructure arrays with adjustable hot spot intensity distribution and reconfigurable position. At the same time, TiO_2_ materials with special properties are introduced to further enhance hot spots. By controlling the etching time of PS colloidal spheres in Ag NC, four different nanostructure arrays are prepared: NC, NCS, NCR, and NPD. After the introduction of TiO_2_, the properties of the four structures are further evaluated, and it is observed that the mixed array of NCR has the strongest SERS signal (etching time is 120 s) and the Raman enhancement factor of SERS reached 4.93 × 10^5^, which is slightly higher than that of the same type of SERS substrate. At the same time, we calculate the charge transfer ability of TiO_2_ in these structures, and the results are in good agreement with the experimental results. FDTD simulation further shows the source of SERS signal is not only the CT process provided by TiO_2_. Large area highly ordered and high-density nanostructures can significantly improve the strength and density of surface plasmon hot spots. This work has potential application value in surface science, chemical detection, nanometer photonics, and so on.

## Figures and Tables

**Figure 1 materials-15-07311-f001:**
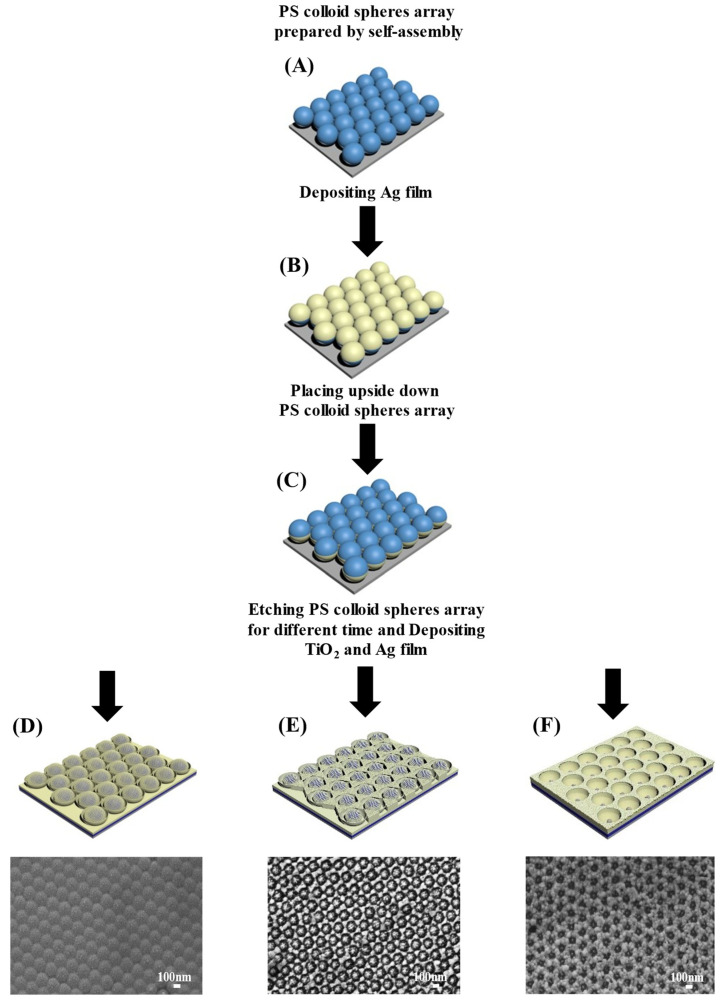
The schematic of preparation of the Ag/TiO_2_/Ag nano structural arrays. (**A**) PS colloid spheres array prepared by self-assembly. (**B**) Depositing Ag film. (**C**) Placing upside down PS colloid spheres array. (**D**–**F**) Etching PS colloid spheres array for different time and depositing TiO_2_ and Ag film.

**Figure 2 materials-15-07311-f002:**
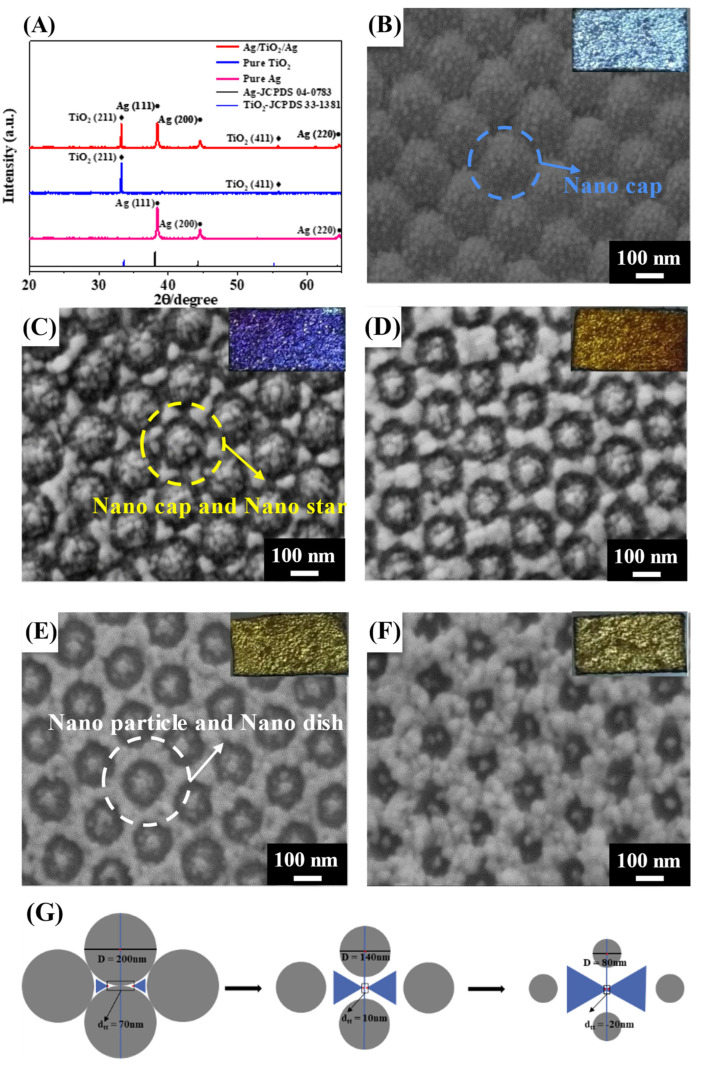
Characterization of composition and morphology of different structures and schematic diagram of their evolution. (**A**) XRD diagram of the Ag/TiO_2_/Ag nano structures, the crystal faces of Ag and TiO_2_ are marked with different labels respectively. (**B**–**F**) The SEM of the Ag/TiO_2_/Ag nano structural arrays. (**B**) Etching 0 s. (**C**) Etching 60 s. (**D**) Etching 120 s. (**E**) Etching 180 s. (**F**) Etching 240 s. (**G**) The evolution of NCS to NCR.

**Figure 3 materials-15-07311-f003:**
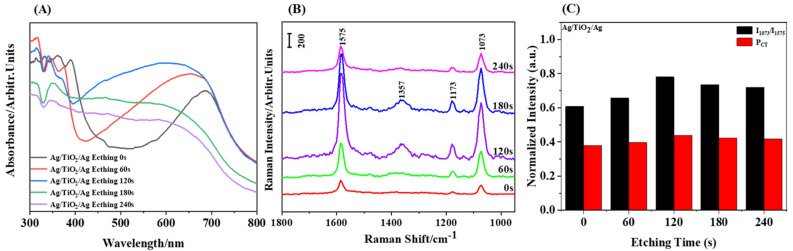
The absorption spectra and SERS spectra of structural change sequences after the intro--duction of TiO_2_. (**A**) The absorption spectra of Ag/Ag nano structural change sequences. (**B**) Ag/TiO_2_/Ag etching 0 s, 60 s, 120 s, 180 s, and 240 s. (**C**) Relative peak intensity and charge transfer for I_1073_/I_1575_ of the Ag/TiO_2_/Ag nanostructure.

**Figure 4 materials-15-07311-f004:**
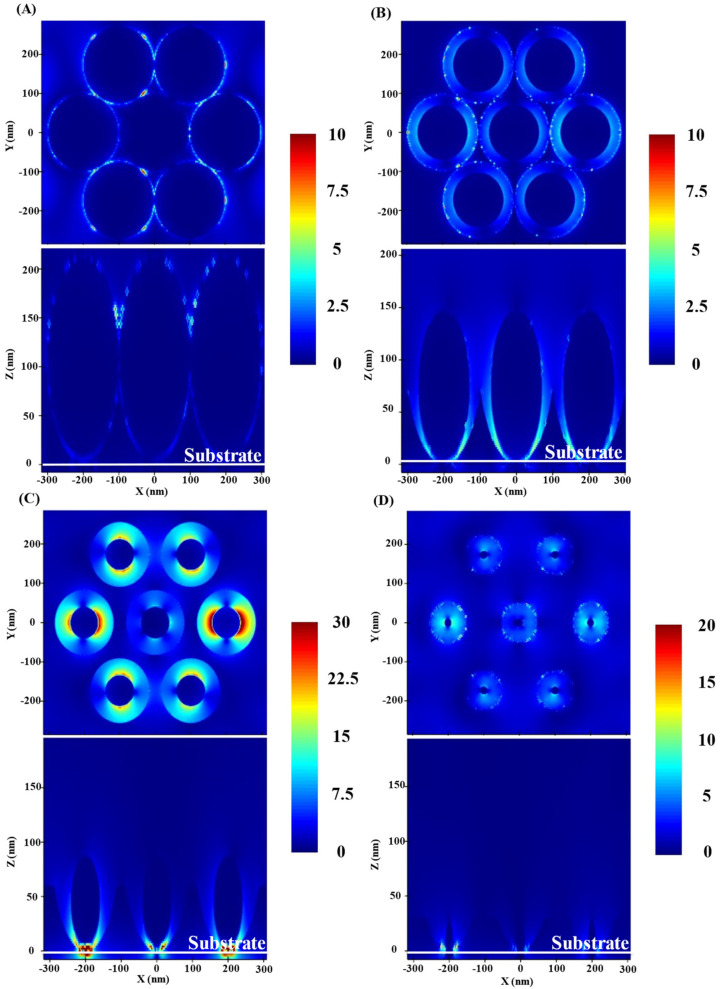
FDTD simulation results of (**A**) NC (etched for 0 s). (**B**) NCS (etched for 60 s). (**C**) NCR (etched for 120 s). (**D**) NPD (etched for 180 s). The top and bottom half of each diagram represent the top and side views of the simulation results, respectively.

## Data Availability

Not applicable.

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
