# Peer review of "Design of Ag/TiO_2_/Ag Composite Nano-Array Structure with Adjustable SERS-Activity"

_materials, 2022, doi:10.3390/ma15207311_

Round 1
Reviewer 1 Report
The manuscript describes the method to prepare, “Design of Ag/TiO2/Ag Composite Nano-Array Structure with Adjustable SERS-activity with developing nanostructure of devices”. Also, he mentions the, nano-etching technology and magnetron sputtering technology are combined to prepare nanostructure substrates with evolvable structure, and Ag/TiO2/Ag composites are introduced into the evolvable composite structure. The structural, and morphological properties of the films were studied. After careful evaluation of the paper, I recommend publication subject to a major revision in the following aspects
1) Revised XRD Figure 2A with (hkl) plans, and added pure TiO2, and Ag particles.
2) Revised the SEM images with better quality with different magnifications.
3) The author should revise induction with the current status of SSERS activity.
4) Abstract should be revised with resultant values.
5) Conclusions should be revised with a scientific aspect.
6) The author should revise the resultant report with the previous report and compere.
Reviewer 2 Report
Journal Materials (ISSN 1996-1944)
Manuscript ID materials-1952460
Title: Design of Ag/TiO2/Ag Composite Nano-Array Structure with Adjustable SERS-activity
Authors: Zhao et al
In this manuscript, authors designed and developed Ag/TiO2/Ag Composite Nano-Array Structures with Adjustable surface-enhanced Raman scattering (SERS)-activity using standard etching process and magnetron sputtering deposition method.
In my opinion, this work is original, and well-presented. I would recommend accept this manuscript for publication in Materials journal after answering following major review comments:
1- Why are the results of this work not compared with other reported results?. I suggest creating a table for comparing absorption spectrum details, and SERS activity in this work with other reported studies.
2- From Figure 3 A, please calculate the bandgap for Ag/TiO2/Ag structures with different etching times. In the manuscript, explain why the bandgap for each structure is different
3- line 226 to 240, you need to add many references for your claims. There are so many references regarding this.
4- please modify the English language through the whole manuscript.
Reviewer 3 Report
Manuscript Number: 1952460 Title: Design of Ag/TiO2/Ag Composite Nano-Array Structure with Adjustable SERS-activity Comments: In this article author synthesized Ag/TiO2/Ag Nano-Array Composite for SERC (surface-enhanced Raman scattering). I recommend major modification needed for this manuscript to be considered. I highlighted some points given below 1. Author should explain how the arrangement of polystyrene colloidal sphere and morphology remains as same after changing the silicon wafer? 2. What is the ratio of Ag and TiO2 in the Ag/TiO2/Ag Nano-Array Composite? 3. Figure quality should need to be improved for Fig.3 graph are tiny so the reader cannot read the data properly and the given information not clearly visible. 4. Author should compare the result with previously reported material with Ag/TiO2/Ag Nano-Array Composite. 5. There are several reports available for SERC (surface-enhanced Raman scattering) so what is the novelty of this work. 6. Author should carefully check the entire manuscript there are some errors in the formula and also please check the grammatical problems. 7. Some review might useful for authors during revision, e.g. 10.1039/D2NJ02481K; 10.1039/D2NA00119E; doi.org/10.1016/j.jallcom.2022.164294; Conducting polymeric nanocomposites: A review in solar fuel applications. Fuel 325 (2022): 124899.
Reviewer 4 Report
In my opinion manuscript materials-1952460 deserves publication after minor revision.
Some suggestions to improve the manuscript:
1. Explain novelty at the end of the abstract.
2. Revise English through the manuscript, for example sentences at lines 25-27 should be united in one phrase.
3. Correct typos, for example: subsection 2.1 subtitle should be “Preparation ….
4. Emphasize originality of the paper at the end of the Introduction.
5. At subsection 2.3 indicate the used Raman laser wavelength, power and the number of scans or acquisition duration.
6. At subsection 2.4. indicate the computational resources used.
7. Part of subsection 3.1 should be moved to Materials and methods section.
8. Explain better the caption of the Fig 1; indicate A, B, … images meaning. Figures should be self-explanatory.
9. Describe insets of the Figure 2 in the Figure caption and give scale bars on the inset pictures.
10. Increase text size of the text in Figures 3 A- C.
11. Compare your results with the performance of other nano array structures.
Round 2
Reviewer 1 Report
I think the authors have revised it well and can be considered for publication from my side.
Reviewer 3 Report
The authors revised well and may considered for publication .